# LOVE: Benchmarking and Evaluating Text-to-Video Generation and Video-to-Text Interpretation

**Jiarui Wang**[1]  **Huiyu Duan**[1]  **Ziheng Jia**[1]  **Zicheng Zhang**[1]  **Yu Zhao**[1]  **Juntong Wang**[1]  **Guangtao Zhai**[1]  **Xiongkuo Min**[1]

## Abstract

Recent advancements in large multimodal models (LMMs) have driven substantial progress in both text-to-video (T2V) generation and video-to-text (V2T) interpretation tasks. However, current AI-generated videos (AIGVs) still exhibit limitations in terms of perceptual quality and text-video alignment. To this end, we present **AIGVE-60K**, a comprehensive dataset and benchmark for AI-Generated Video Evaluation, which features **(i) comprehensive tasks**, encompassing 3,050 extensive prompts across 20 fine-grained task dimensions, **(ii) the largest human annotations**, including 120K mean-opinion scores (MOSs) and 60K question-answering (QA) pairs annotated on 58,500 videos generated from 30 T2V models, and **(iii) bidirectional benchmarking and evaluating** for both T2V generation and V2T interpretation capabilities. Based on AIGVE-60K, we propose **LOVE**, a LMM-based metric for AIGV Evaluation from multiple dimensions including perceptual preference, text-video correspondence, and task-specific accuracy. Building upon LOVE, we further introduce **LOVE-Reward** to optimize T2V models through reinforcement learning, effectively enhancing both the perceptual quality and text-video correspondence of generated videos. Comprehensive experiments demonstrate that LOVE achieves state-of-the-art performance and generalizes effectively to various AIGV benchmarks. LOVE-Reward significantly improves video generation quality. These findings highlight the effectiveness of the AIGVE-60K dataset and our proposed methods. The database and codes are available at https://github.com/IntMeGroup/LOVE.

[1]Institute of Image Communication and Network Engineering, Shanghai Jiao Tong University, Shanghai, China. Correspondence to: Xiongkuo Min <minxiongkuo@sjtu.edu.cn>.

## 1. Introduction

The rapid advancement of large multimodal models (LMMs) has revolutionized both text-to-video (T2V) generation (Singh, 2023; Xing et al., 2024; Liao et al., 2024) and video-to-text (V2T) interpretation (Zhang et al., 2023; Ye et al., 2024; Li et al., 2024c), leading to high-quality video generation and comprehensive multimodal video understanding capabilities. However, state-of-the-art T2V models may still produce videos with degraded **perceptual quality** and limited **text-video correspondence**, thus may fail to meet human preferences (Wang et al., 2025b; Li et al., 2024a; Huang et al., 2024). Given the high cost and inefficiency of human evaluation, it is of great significance to develop a reliable and scalable evaluation metric that aligns well with human preferences for AI-generated videos (AIGVs).

To fairly and effectively evaluate T2V models and AIGVs, many T2V model benchmarks and AIGV evaluation datasets (Zheng et al., 2025; Zhang et al., 2024; Kou et al., 2024) have been constructed as shown in Table 1, and many AIGV evaluation metrics have been proposed (Li et al., 2019; Sun et al., 2022; Wu et al., 2022; 2023a). However, these efforts face the following limitations that may affect their effectiveness in diverse applications. (1) **Most benchmarks or datasets only consider either perception, correspondence, or task-specific accuracy dimensions, while comprehensive subjective evaluation works are still lacking.** Since high-quality AIGVs may exhibit poor text-video alignment, well-aligned AIGVs may suffer from low perceptual quality (Wang et al., 2025b), and we also need a binary true-or-false metric in some scenarios (such as number-based generation scenes) (Ghosh et al., 2023; Zheng et al., 2025), an extensive evaluation is important. However, some existing metrics only focus on one dimension (Li et al., 2024a) or use the fused overall evaluation (Liu et al., 2024c; Kou et al., 2024). (2) **The scale of the datasets remains small and annotations remain coarse.** Some datasets or benchmarks include only a limited number of T2V models (Zheng et al., 2025; Liu et al., 2024c; Li et al., 2024a) or AIGVs (Chivileva et al., 2023; Zhang et al., 2024; Kou et al., 2024), which constrains the ability to validate the effectiveness and scalability of evaluation methods across diverse

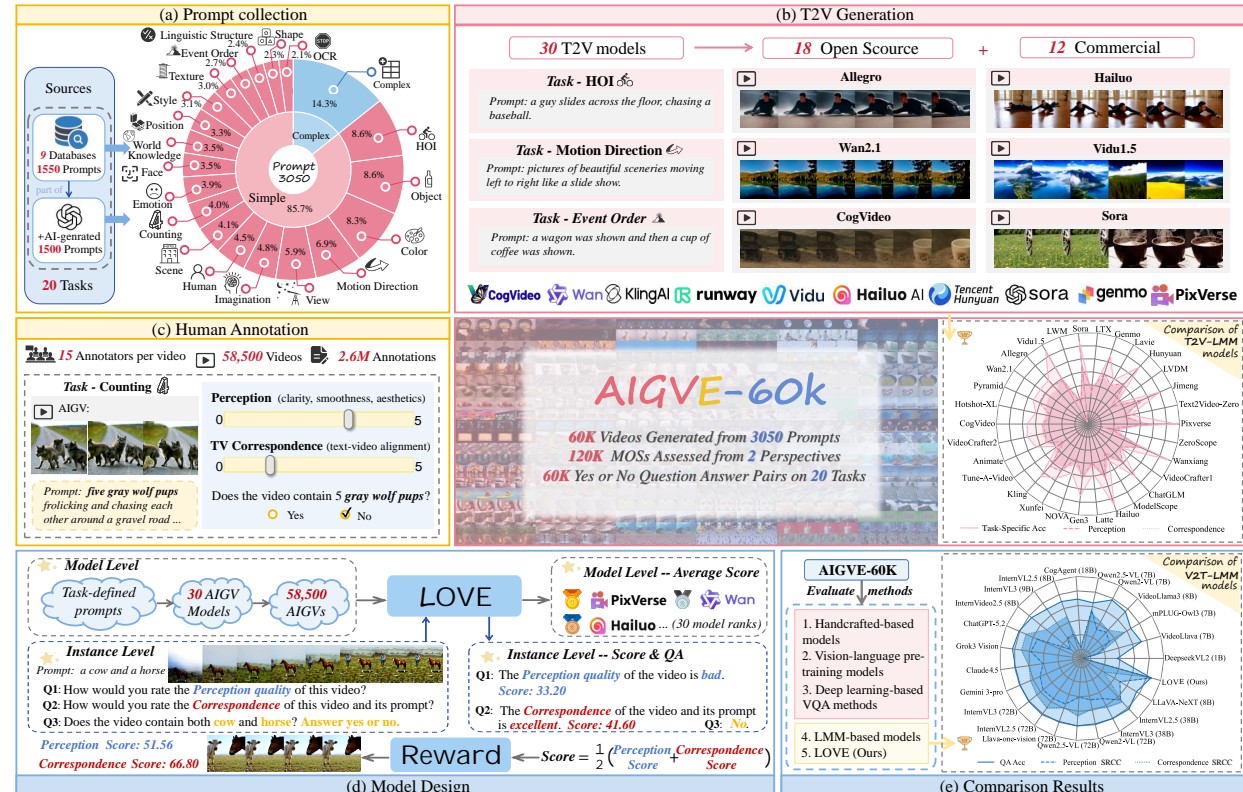

*Figure 1.* We present the largest AIGV evaluation database (**AIGVE-60K**) and a novel model (**LOVE**). (a) We collect 3,050 prompts across 20 fine-grained tasks. (b) 30 T2V models are applied to generate 60K videos. (c) 120K MOSs from 2 perspectives and 60K question-answer pairs are acquired from annotators. (d) We design **LOVE** model to evaluate **T2V generation** at both the instance and model levels and **LOVE-Reward** approach to improve T2V generation. (e) Comparison results of LMM's **V2T interpretation** ability.

models and outputs, and some works use coarse-MOSs (Liu et al., 2024b; Zhang et al., 2025b), which do not meet ITU standards (Series, 2012) and may produce invalid MOSs. (3) **Current evaluation metrics for T2V generation consider from either the instance level or the model level, while a comprehensive study integrates both perspectives remains lacking.** For example, Inception Score (IS) (Gulrajani et al., 2017), Fréchet Video Distance (FVD) (Unterthiner et al., 2018), VBench (Huang et al., 2024), *etc.*, are mainly designed to evaluate T2V models, *i.e.*, from a model level perspective, while LGVQ (Zhang et al., 2024), AIGV-Assessor (Wang et al., 2025b), *etc.*, are mainly designed for the individual AIGV evaluation or multiple AIGV comparison, *i.e.*, from a instance level perspective. To the best of our knowledge, no existing work has jointly considered both perspectives towards a comprehensive evaluation.

To address these challenges, we present **AIGVE-60K**, a large-scale dataset and benchmark for AI-Generated Video Evaluation, which includes 58,500 videos generated by 30 state-of-the-art T2V models using 3,050 diverse prompts across 20 task-specific challenges. As shown in Figure 1, we collect **2.6M** human annotations from the perception, text-video correspondence, and task-specific accuracy, respectively, and finally obtain 120K mean opinion scores (MOSs) and 60K question-answering (QA) pairs. Based on

AIGVE-60K, we propose **LOVE**, a LMM-based metric for AIGV Evaluation from multiple dimensions at both instance level and model level, which integrates: (1) dual encoders for vision-temporal feature extraction, (2) a large-language model (LLM) backbone for ***all-in-one*** video quality assessment, and (3) instruction tuning techniques (Liu et al., 2023) for accurate response generation. Through extensive experimental validation, we demonstrate that LOVE achieves state-of-the-art performance on the AIGVE-60K dataset and manifests strong zero-shot generalization ability on other benchmarks. Building upon the LOVE model, we further introduce **LOVE-Reward**, which employs LOVE as a reward function to directly fine-tune text-to-video generation models. This approach effectively optimizes the generation process, leading to significant improvements in both the perceptual quality and text-video correspondence of the generated videos. Our contributions are summarized as follows:

- We present **AIGVE-60K**, **the largest text-to-video evaluation dataset** that contains 58,500 generated videos with 2.6M subjective ratings from the perception, text-video correspondence, and task-specific accuracy, respectively.

- We introduce **a bidirectional benchmarking** and evaluation strategy. We benchmark the T2V *generation*

*Table 1.* Comparison of T2V model evaluation benchmarks and AIGV quality evaluation databases.

| Database | Annotation Type (People per Sample) | Videos | Prompts | Annotations | Models | Evaluation Concern | | | |
|---|---|---|---|---|---|---|---|---|---|
| | | | | | | T2V Tasks | Perception | T2V Correspondence | QA Acc |
| VBench (Huang et al., 2024) | Pairs (1) | 3,200 | 800 | 24,000 | 4 | 16 | | ✓ | ✗ |
| VBench2.0 (Zheng et al., 2025) | Pairs (1) | 100,800 | 1,260 | 151,200 | 4 | 18 | ✗ | ✓ | ✓ |
| FETV (Liu et al., 2024c) | Coarse-MOS (3) | 2,476 | 619 | 7,428 | 4 | 5 | | *Overall* | ✗ |
| GenAIBench (Li et al., 2024a) | Coarse-MOS (3) | 32,000 | 800 | 9,600 | 4 | 8 | ✗ | ✓ | ✗ |
| Q-Eval (Zhang et al., 2025b) | Coarse-MOS (3) | 40,000 | 2,500 | 384,000 | 16 | ✗ | ✓ | ✓ | ✓ |
| MQT (Chivileva et al., 2023) | Fine-MOS (24) | 1,005 | 201 | 48,240 | 5 | ✗ | ✓ | ✓ | ✓ |
| LGVQ (Zhang et al., 2024) | Fine-MOS (20) | 2,808 | 468 | 168,480 | 6 | ✗ | ✓ | ✓ | ✗ |
| T2VQA-DB (Kou et al., 2024) | Fine-MOS (27) | 10,000 | 1,000 | 270,000 | 9 | ✗ | | *Overall* | ✗ |
| AIGVQA-DB (Wang et al., 2025b) | Fine-MOS (20) & Pairs (3) | 36,576 | 2,048 | 371,520 | 15 | ✗ | ✓ | ✓ | ✓ |
| **AIGVE-60K (Ours)** | **Fine-MOS (15)** | **58,500** | **3,050** | **2,632,500** | **30** | **20** | ✓ | ✓ | ✓ |

*ability* of 30 T2V models, and the V2T *interpretation ability* of 23 LMMs and 24 VQA metrics.

- We propose **LOVE**, **a novel LMM-based evaluation model** capable of assessing both the perceptual quality and T2V alignment for AIGVs. Experimental results on diverse benchmarks manifest the state-of-the-art performance and strong generalization ability of LOVE.

- We introduce **LOVE-Reward**, for **optimizing the text-to-video generation process**, which significantly enhances the quality of the generated videos.

## 2. Related Works

### 2.1. Benchmarks for T2V Generation

As shown in Table 1, current T2V model evaluation benchmarks and VQA databases can be categorized into pairs, coarse MOS, and fine-MOS based on the annotation method and granularity. VBench (Huang et al., 2024) and VBench2.0 (Zheng et al., 2025) focus on video pairs comparison, but are limited in T2V comparison model number and lack precise quality assessment for each AIGV. Fine-MOS databases offer more reliable assessments derived from more than 15 annotators, following the guidelines of ITU-R BT.500 (Series, 2012). MQT (Chivileva et al., 2023), LGVQ (Zhang et al., 2024) collect fine-grained MOSs but the number of AIGVs is limited. AIGVQA-DB (Wang et al., 2025b) considers both perceptual quality and T2V correspondence, however, it mainly focuses on the pair comparison. AIGVE-60K stands out by its largest scale of annotations, providing fine-grained MOSs and answer annotations for task-specific questions.

### 2.2. Evaluation Metrics for T2V Generation

Many quality assessment models have been proposed in the literature (Mittal et al., 2012b; Li et al., 2022a; Kirstain et al., 2023; Li et al., 2024a), including handcrafted models (*e.g.*, QAC (Xue et al., 2013), BRISQUE (Mittal et al., 2012a)) and deep learning-based VQA models (*e.g.*, FAST-VQA (Wu et al., 2022), DOVER (Wu et al., 2023a)). These models characterize quality-aware information to predict perception quality scores but can not evaluate T2V correspondence. PickScore (Kirstain et al., 2023) and VQAScore (Li et al., 2024a) improve the evaluation of the T2V correspondence,

but they struggle to assess the perception quality of AIGV. VBench (Huang et al., 2024) employs various detection models for task-specific accuracy, but this approach is quite complex. Our proposed LMM-based model complies with an ***all-in-one*** framework, which can evaluate quality scores and task-specific accuracies in one model.

## 3. AIGVE-60K Dataset & Benchmark

In this section, we introduce the construction of AIGVE-60K and **benchmark T2V models** based on the dataset.

### 3.1. Data Collection

Prompts of the AIGVE-60K are primarily sourced from 9 existing open-domain text-video pair datasets and some are refined using DeepSeek R1 (Guo et al., 2025) to expand and modify them, ensuring clarity and diversity. Our prompt design focuses on 20 different tasks as shown in Figure 1(a). The complex tasks are designed by combining simpler task components, such as motion direction, event order, and counting, into more complex challenges. In total, we collect 3,050 prompts, each corresponding to a specific task. To generate the AIGVs, we utilize 30 of the latest T2V models, as shown in Figure 1(b). We leverage open-source website APIs or the default weights of these models to generate videos. For the training set, we employ 18 open-source models and generate 49,500 videos. The test set includes 9,000 video generated by 30 models. With 3,050 distinct prompts, this process results in a total of 58,500 videos.

### 3.2. Subjective Experiment Setup and Procedure

Due to the unique distortions in AIGVs and varying elements determined by different text prompts, relying solely on an overall score for evaluation is inadequate. In this paper, we propose to evaluate AIGVs across two dimensions. (1) **Perceptual quality** focuses on visual perception, evaluating factors such as detail richness, motion smoothness, color vibrancy, and distortion levels. (2) **Text-video correspondence** evaluates how accurately the generated video reflects the objects, scenes, styles, and details described in the text prompt. We use a 1-5 Likert scale to score the videos based on the perception and T2V correspondence. For the correspondence evaluation, in addition to the rating, annotators are instructed to answer task-specific yes/no questions to determine whether the video consistently aligns

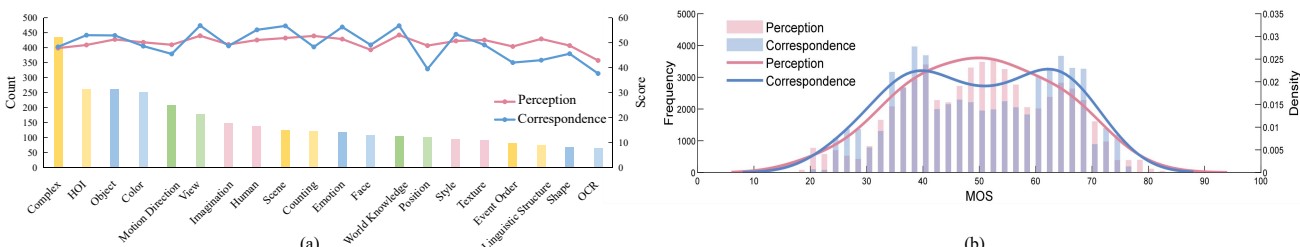

*Figure 2.* (a) Distribution of task counts and scores across different tasks. (b) Distribution of perception and correspondence MOSs.

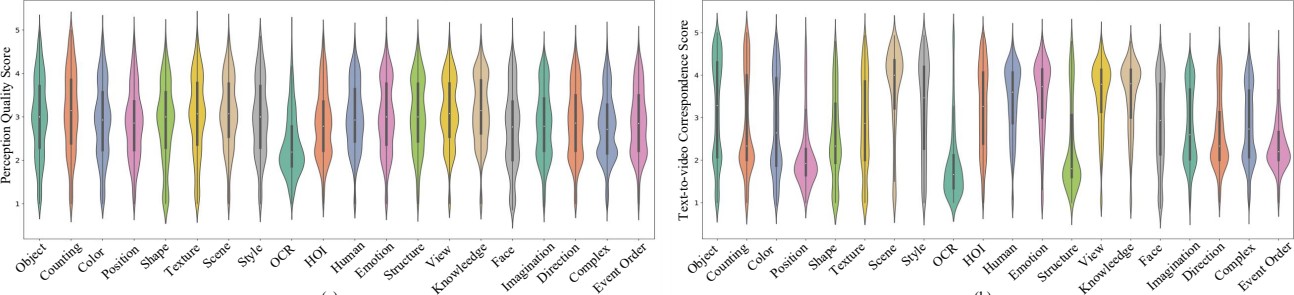

*Figure 3.* The MOS distribution in terms of different task contents: (a) perception quality (b) T2V correspondence.

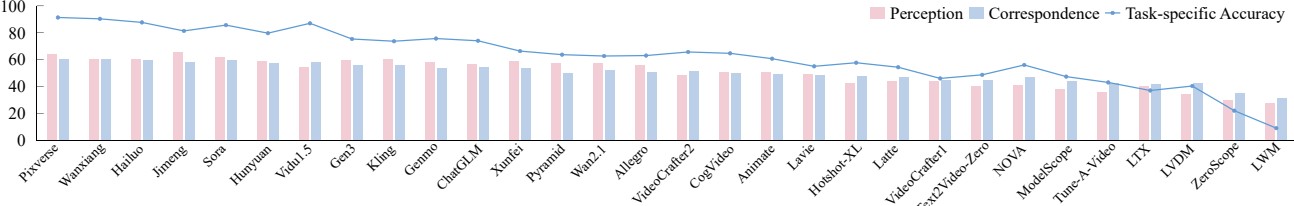

*Figure 4.* Comparison of T2V generation models regarding the perception MOSs, correspondence MOSs, and task-specific accuracy.

with the prompt. Finally, we obtain a total of 2,632,500 human annotations including 1,755,000 reliable score ratings (15 annotators $\times$ 2 dimensions $\times$ 58,500 videos), and 877,500 task-specific QA pairs. In order to obtain the MOS for an AIGV, we first convert the raw ratings into Z-scores, and then linearly scale them to the range $[0, 100]$ as follows:

$$z_{ij} = \frac{r_{ij} - \mu_i}{\sigma_i}, \quad z'_{ij} = \frac{100(z_{ij} + 3)}{6}, \quad (1)$$

$$\mu_i = \frac{1}{N_i} \sum_{j=1}^{N_i} r_{ij}, \quad \sigma_i = \sqrt{\frac{1}{N_i - 1} \sum_{j=1}^{N_i} (r_{ij} - \mu_i)^2}, \quad (2)$$

where $r_{ij}$ is the raw rating given by the $i$-th subject to the $j$-th video. $N_i$ is the number of videos judged by subject $i$. Next, the MOS of the $j$-th video is computed by averaging the rescaled z-scores across all subjects. The task-specific yes/no answer is determined by the most votes. A total of 117,000 MOSs (2 dimensions $\times$ 58,500 videos) and 58,500 question answering pairs are obtained.

### 3.3. Data Analysis & T2V Model Benchmark

The distribution of task counts and averaged scores is shown in Figure 2(a). It can be observed that the correspondence score fluctuates more than the quality score, which means that the text-video alignment is more sensitive to the tasks.

The distribution of MOSs for both T2V correspondence and perceptual quality is shown in Figure 2(b), which approximately follows the Gaussian distributions. Figure 3 displays the MOS distribution for each task, demonstrating notable differences in model capabilities between tasks, especially for T2V correspondence. Moreover, we also launch comparisons for T2V generation models across different tasks based on perceptual quality MOSs, correspondence MOSs, and task-specific accuracy, as shown in Figure 4.

## 4. The LOVE and LOVE-Reward Method

In this section, we present an ***all-in-one*** video quality assessment method **LOVE** to identify quality degradation levels, predict perception and correspondence scores, and deliver visual question answers within a unified model. Building upon LOVE, we further introduce **LOVE-Reward**, a reinforcement learning framework that leverages LOVE's multi-dimensional evaluation to optimize T2V generation models, enhancing both perceptual quality and text-video correspondence through reward-guided fine-tuning.

### 4.1. LOVE Model Structure and Training Strategy

**Model Structure.** Figure 5 illustrates that the visual encoding component comprises a vision encoder and a temporal encoder for feature extraction. The vision encoder is

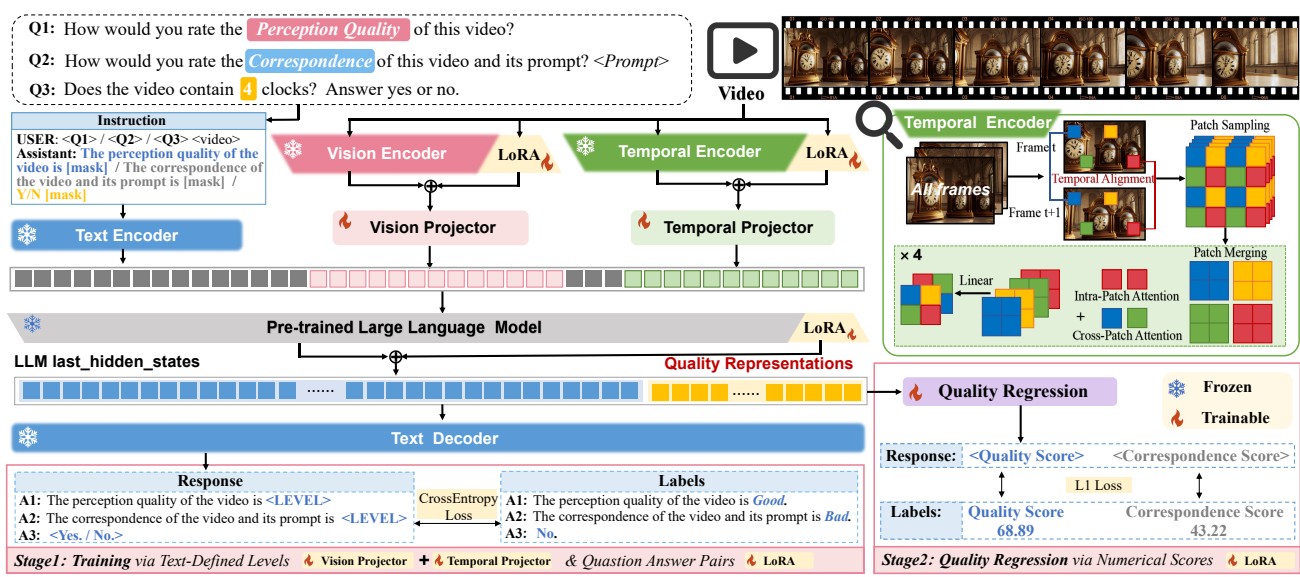

*Figure 5.* Overview of the LOVE architecture. The model includes two functions: (1) text-defined quality level and score prediction, (2) task-specific visual question answering. The training process consists of two stages: training via text-defined levels, and quality regression via numerical scores. The model incorporates a vision encoder, a temporal encoder, and a text encoder for extracting visual and textual features, which are fed into a pre-trained LLM to generate results. LoRA (Hu et al., 2022) weights are introduced to the image encoder and LLM to adapt the models to perception quality and T2V correspondence evaluation, and 20 task-specific visual question-answering.

constructed upon a pre-trained vision transformer (ViT), *i.e.*, InternViT (Chen et al., 2024b). For the temporal encoder, we decompose all video frames $\{F^{(n)}\}_{n=1}^{N_v}$ into temporally aligned mini-patch map through grid mini-patch sampling. For each video frame $F^{(n)}$, we first split it into uniform $L \times L$ grids, the set of grids $G^{(n)}$ can be described as:

$$G^{(n)} = \{g_{0,0}^{(n)}, \cdots, g_{i,j}^{(n)}, \cdots, g_{L,L}^{(n)}\}, \tag{3}$$

$$g_{i,j}^{(n)} = F^{(n)}[\frac{i \times H}{L} : \frac{(i+1) \times H}{L}, \frac{j \times W}{L} : \frac{(j+1) \times W}{L}] \tag{4}$$

where $g_{i,j}^{(n)} \in \mathbb{R}^{\frac{H}{L} \times \frac{W}{L} \times 3}$ denotes the grid in the $i$-th row and $j$-th column of $F^{(n)}$. We sample the mini-patches from each $g_{i,j}^{(n)}$ and splice all the selected mini-patches to get the mini-patch map $M \in \mathbb{R}^{H \times W \times 3}$, which are then fed into a Swin-T (Liu et al., 2021) with four hierarchical self-attention layers. To align the extracted features with the input space of the LLM, a vision projector and a temporal projector with two multilayer perceptron (MLP) layers are applied. We utilize the InternLM3-9B-instruct (Wang et al., 2024) to integrate the visual tokens and text instruction tokens to perform the following two tasks: (1) produces an evaluation of the input video's quality level, such as "*The perception quality of the video is (bad, poor, fair, good, excellent).*" (2) takes the quality representations from the last hidden states of the LLM to perform regression through a quality regression module, outputting numerical scores.

**Training and Fine-tuning Strategy.** LOVE is trained using a two-stage methodology. We first perform instruction tuning via text-defined quality levels and then train the quality regression module via numerical scores with LoRA (Hu et al., 2022). We first convert the continuous scores into categorical text-based quality levels. Specifically, we uniformly divide the range between the highest score (M) and the lowest score (m) into five intervals:

$$L(s) = l_V \text{ if m} + \frac{i-1}{5} \times (\text{M}-\text{m}) < s \le \text{m} + \frac{i}{5} \times (\text{M}-\text{m}), \tag{5}$$

where $\{l_V|_{i=1}^5\} = \{$*bad, poor, fair, good, excellent*$\}$ are the standard text rating levels as defined by ITU (Series, 2012). We then take the last-hidden-state features to a quality regression module to generate more accurate quality scores. We use standard language loss during the instruction tuning phase and employ L1 loss for the quality regression task.

### 4.2. The LOVE-Reward Strategy

To enhance the quality of generated videos while maintaining training efficiency, we propose **LOVE-Reward**, which leverages the LOVE metric as a reward function to optimize text-to-video generation models through reinforcement learning. We adopt the GRPO (Liu et al., 2025; Shao et al., 2024) with the following objective function:

$$\mathcal{J}_{\text{LOVE-Reward}}(\theta) = \mathbb{E}_{\{\boldsymbol{x}^i\}_{i=1}^G \sim \pi_{\theta_{\text{old}}}} \left[ \frac{1}{G} \sum_{i=1}^{G} \frac{1}{T} \sum_{t=0}^{T-1} L_t^i(\theta, \varepsilon, \beta) \right], \tag{6}$$

$$L_t^i(\theta, \varepsilon, \beta) = \min\left(r_t^i(\theta)\hat{A}^i, \text{ clip}\left(r_t^i(\theta), 1-\varepsilon, 1+\varepsilon\right)\hat{A}^i\right)$$

$$-\beta D_{\text{KL}}(\pi_\theta \| \pi_{\text{ref}}), r_t^i(\theta) = \frac{p_\theta(\boldsymbol{x}_{t-1}^i \mid \boldsymbol{x}_t^i, \boldsymbol{c})}{p_{\theta_{\text{old}}}(\boldsymbol{x}_{t-1}^i \mid \boldsymbol{x}_t^i, \boldsymbol{c})}. \tag{7}$$

*Table 2.* **Performance benchmark on AIGVE-60K.** *Overall-Averaged* indicates that we average the MOS of the two dimensions as the overall video quality score. ♠Conventional handcrafted metrics, ◇VBench metrics, ♣deep learning-based VQA models, ♡vision-language pre-training models, ★open-source LMM-based models, and △close-source LMM-based models. ♦*Refers to fine-tuned models. The best results are marked in **RED** and the second-best in **BLUE**.

| Dimension | Perception Quality | | | | T2V Correspondence | | | | Overall-Averaged | | | | Question Answer | |
|---|---|---|---|---|---|---|---|---|---|---|---|---|---|---|
| Evaluation Level | Instance-level | | Model-level | | Instance-level | | Model-level | | Instance-level | | Model-level | | Instance | Model |
| Methods / Metrics | SRCC↑ | PLCC↑ | SRCC↑ | PLCC↑ | SRCC↑ | PLCC↑ | SRCC↑ | PLCC↑ | SRCC↑ | PLCC↑ | SRCC↑ | PLCC↑ | Acc (%)↑ | SRCC↑ |
| ♠BMPRI (Min et al., 2018) | 0.5741 | 0.4976 | 0.7878 | 0.7312 | 0.3618 | 0.3214 | 0.7321 | 0.6582 | 0.5106 | 0.4452 | 0.7709 | 0.7112 | 64.00 | 0.7023 |
| ♠BPRI (Min et al., 2017) | 0.3558 | 0.3403 | 0.6356 | 0.7122 | 0.2018 | 0.2012 | 0.5324 | 0.6256 | 0.3044 | 0.2945 | 0.5800 | 0.6863 | 63.56 | 0.4701 |
| ♠BRISQUE (Mittal et al., 2012a) | 0.5843 | 0.4881 | 0.8131 | 0.7196 | 0.3806 | 0.3325 | 0.7615 | 0.6598 | 0.5239 | 0.4459 | 0.8011 | 0.7048 | 64.67 | 0.7366 |
| ♠QAC (Xue et al., 2013) | 0.5958 | 0.5717 | 0.8100 | 0.7447 | 0.3948 | 0.3430 | 0.7717 | 0.6557 | 0.5421 | 0.4974 | 0.8029 | 0.7182 | 64.40 | 0.7495 |
| ◇V-Aesthetic Quality (Huang et al., 2024) | 0.5031 | 0.5315 | 0.7740 | 0.8309 | 0.4033 | 0.4358 | 0.7273 | 0.7986 | 0.4877 | 0.5254 | 0.7526 | 0.8291 | 64.54 | 0.7046 |
| ◇V-Imaging Quality (Huang et al., 2024) | 0.2810 | 0.3093 | 0.5426 | 0.4766 | 0.1952 | 0.2244 | 0.4986 | 0.5101 | 0.2531 | 0.2900 | 0.5186 | 0.4970 | 60.60 | 0.4389 |
| ◇V-Overall Consistency (Huang et al., 2024) | 0.1559 | 0.1939 | 0.1742 | 0.3319 | 0.3076 | 0.3685 | 0.3201 | 0.5419 | 0.2510 | 0.3046 | 0.2338 | 0.4233 | 61.96 | 0.3353 |
| ◇V-Subject Consistency (Huang et al., 2024) | 0.3443 | 0.3968 | 0.4839 | 0.6065 | 0.1647 | 0.2425 | 0.4416 | 0.5744 | 0.2739 | 0.3476 | 0.4679 | 0.6016 | 62.52 | 0.4545 |
| ♣VSFA (Li et al., 2019) | 0.3750 | 0.3898 | 0.6227 | 0.5987 | 0.2438 | 0.2645 | 0.5858 | 0.5777 | 0.3320 | 0.3556 | 0.6102 | 0.5983 | 57.09 | 0.5204 |
| ♣BVQA (Li et al., 2022a) | 0.3089 | 0.3512 | 0.5030 | 0.6506 | 0.2379 | 0.2675 | 0.4674 | 0.6009 | 0.3023 | 0.3361 | 0.4897 | 0.6390 | 58.47 | 0.4781 |
| ♣SimpleVQA (Sun et al., 2022) | 0.5631 | 0.5670 | 0.8038 | 0.7873 | 0.3474 | 0.3487 | 0.7273 | 0.7327 | 0.4884 | 0.4979 | 0.7620 | 0.7756 | 60.78 | 0.6894 |
| ♣FAST-VQA (Wu et al., 2022) | 0.6391 | 0.6394 | 0.8945 | 0.9063 | 0.3919 | 0.4162 | 0.8376 | 0.8492 | 0.5563 | 0.5738 | 0.8683 | 0.8952 | 66.27 | 0.8122 |
| ♣DOVER (Wu et al., 2023a) | 0.6414 | 0.6552 | 0.8874 | 0.9244 | 0.3759 | 0.3936 | 0.8038 | 0.8448 | 0.5480 | 0.5702 | 0.8496 | 0.9043 | 62.61 | 0.7662 |
| ♡CLIPScore (Hessel et al., 2021) | 0.0947 | 0.1348 | 0.0300 | 0.1732 | 0.2290 | 0.2818 | 0.1408 | 0.3835 | 0.1738 | 0.2256 | 0.0848 | 0.2625 | 58.27 | 0.1695 |
| ♡BLIPScore (Li et al., 2022b) | 0.1884 | 0.2341 | 0.2111 | 0.4336 | 0.3163 | 0.3813 | 0.3451 | 0.5354 | 0.2735 | 0.3335 | 0.2814 | 0.4277 | 63.93 | 0.3775 |
| ♡ImageReward (Xu et al., 2023) | 0.4180 | 0.4472 | 0.8016 | 0.8097 | 0.5076 | 0.5419 | 0.8549 | 0.8790 | 0.4992 | 0.5365 | 0.8331 | 0.8495 | 68.33 | 0.8586 |
| ♡PickScore (Kirstain et al., 2023) | 0.4026 | 0.4193 | 0.8198 | 0.8344 | 0.4135 | 0.4246 | 0.7775 | 0.8497 | 0.4395 | 0.4580 | 0.8033 | 0.8522 | 62.29 | 0.7844 |
| ♡HPSv2 (Wu et al., 2023c) | 0.5415 | 0.5690 | 0.7504 | 0.8096 | 0.4989 | 0.5325 | 0.7522 | 0.8379 | 0.5605 | 0.5980 | 0.7468 | 0.8325 | 67.68 | 0.7789 |
| ♡VQAScore (Li et al., 2024a) | 0.1677 | 0.1961 | 0.3437 | 0.3365 | 0.1763 | 0.2049 | 0.3922 | 0.4173 | 0.1880 | 0.2176 | 0.3811 | 0.3746 | 52.97 | 0.3280 |
| ★VideoLlava (7B) (Lin et al., 2023) | 0.1809 | 0.1938 | 0.6125 | 0.6817 | 0.2005 | 0.2127 | 0.6406 | 0.6700 | 0.2026 | 0.2267 | 0.6764 | 0.7505 | 68.46 | 0.7548 |
| ★mPLUG-Owl3 (7B) (Ye et al., 2024) | 0.3532 | 0.3542 | 0.7962 | 0.7789 | 0.5478 | 0.5551 | 0.9310 | 0.9273 | 0.5540 | 0.5626 | 0.9008 | 0.8886 | 63.02 | 0.8897 |
| ★VideoLlama3 (8B) (Zhang et al., 2025a) | 0.3922 | 0.4196 | 0.9073 | 0.9144 | 0.4228 | 0.4817 | 0.9075 | 0.8569 | 0.5063 | 0.5432 | 0.9576 | 0.9550 | 70.16 | 0.8244 |
| ★Qwen2.5-VL (7B) (Bai et al., 2025) | 0.5410 | 0.5410 | 0.8652 | 0.8683 | 0.5110 | 0.5223 | 0.8167 | 0.8692 | 0.5578 | 0.5799 | 0.8888 | 0.8949 | 62.34 | 0.6655 |
| ★InternVL3 (9B) (Wang et al., 2024) | 0.2731 | 0.3519 | 0.8300 | 0.7576 | 0.4768 | 0.5334 | 0.9373 | 0.9610 | 0.4793 | 0.5307 | 0.9270 | 0.9224 | 65.82 | 0.7719 |
| ★InternVideo2.5 (8B) (Wang et al., 2025c) | 0.1563 | 0.5454 | 0.3361 | 0.6574 | 0.4978 | 0.5538 | 0.9560 | 0.9646 | 0.4430 | 0.5366 | 0.9079 | 0.8968 | 70.64 | 0.8435 |
| ★InternVL3 (38B) (Wang et al., 2024) | 0.4950 | 0.5150 | 0.8118 | 0.8082 | 0.5996 | 0.5920 | 0.9439 | 0.9463 | 0.6040 | 0.5984 | 0.9128 | 0.8946 | 73.89 | 0.9386 |
| ★Qwen2.5-VL (72B) (Bai et al., 2025) | 0.4245 | 0.4562 | 0.7762 | 0.7602 | 0.6272 | 0.5884 | 0.9364 | 0.9522 | 0.5891 | 0.5794 | 0.9239 | 0.9083 | 73.83 | 0.9516 |
| ★Llava-one-vision (72B) (Li et al., 2024b) | 0.5291 | 0.5196 | 0.7829 | 0.7689 | 0.5702 | 0.5510 | 0.8741 | 0.9055 | 0.6010 | 0.5827 | 0.8478 | 0.8497 | 73.31 | 0.9112 |
| ★InternVL3 (72B) (Wang et al., 2024) | 0.5441 | 0.4973 | 0.8923 | 0.8131 | 0.6314 | 0.6055 | 0.9444 | 0.9529 | 0.6405 | 0.6047 | 0.9212 | 0.9014 | 74.59 | 0.9623 |
| △Gemini3-pro (Team, 2026b) | 0.4972 | 0.5012 | 0.8790 | 0.7965 | 0.6095 | 0.5874 | 0.9430 | 0.9491 | 0.5957 | 0.5985 | 0.9346 | 0.9196 | 73.38 | 0.9512 |
| △Claude4.5 (Team, 2026a) | 0.4267 | 0.4711 | 0.7602 | 0.7343 | 0.5827 | 0.5809 | 0.8919 | 0.9238 | 0.5532 | 0.5822 | 0.8598 | 0.8481 | 73.20 | 0.9395 |
| △Grok Vision (xAI Team, 2026) | 0.5628 | 0.5728 | 0.8808 | 0.8403 | 0.6659 | 0.6730 | 0.9546 | 0.9763 | 0.6629 | 0.6784 | 0.9399 | 0.9368 | 76.51 | 0.9469 |
| △GPT-5.2 (Team, 2026c) | 0.5263 | 0.5233 | 0.9048 | 0.8905 | 0.6639 | 0.6343 | 0.9458 | 0.9555 | 0.6240 | 0.6117 | 0.9346 | 0.9326 | 74.84 | 0.9317 |
| ♦InternVL3* (9B) (Wang et al., 2024) | 0.6421 | 0.6575 | 0.9061 | 0.9426 | 0.5965 | 0.6115 | 0.9671 | 0.9659 | 0.6571 | 0.6926 | 0.9341 | 0.9597 | 78.36 | 0.9754 |
| ♦Qwen2.5-VL* (7B) (Bai et al., 2025) | 0.7868 | 0.7996 | 0.9265 | 0.9510 | 0.7354 | 0.7493 | 0.9634 | 0.9677 | 0.7798 | 0.8046 | 0.9579 | 0.9716 | 77.35 | 0.9698 |
| ♦InternVideo2.5* (8B) (Wang et al., 2025c) | 0.7845 | 0.8102 | 0.9308 | 0.9605 | 0.6773 | 0.6946 | 0.9608 | 0.9679 | 0.7802 | 0.8094 | 0.9542 | 0.9721 | 73.37 | 0.9136 |
| ♦LOVE (Ours) | 0.7932 | 0.8259 | 0.9324 | 0.9725 | 0.7466 | 0.7657 | 0.9778 | 0.9825 | 0.8115 | 0.8361 | 0.9657 | 0.9779 | 78.69 | 0.9769 |

To estimate the advantage, for each prompt $c$ we sample a group of $G$ video trajectories and compute the normalized advantage:

$$\hat{A}^i = \frac{R(\boldsymbol{x}_0^i, \boldsymbol{c}) - \text{mean}(\{R(\boldsymbol{x}_0^i, \boldsymbol{c})\}_{i=1}^G)}{\text{std}(\{R(\boldsymbol{x}_0^i, \boldsymbol{c})\}_{i=1}^G)}, \qquad (8)$$

where the reward function $R(\boldsymbol{x}, \boldsymbol{c})$ combines perceptual quality and text-video alignment scores from LOVE:

$$R(\boldsymbol{x}, \boldsymbol{c}) = 0.5 \cdot \text{Perception}(\boldsymbol{x}, \boldsymbol{c}) + 0.5 \cdot \text{Correspondence}(\boldsymbol{x}, \boldsymbol{c}). \qquad (9)$$

## 5. Experiments

In this section, we conduct extensive experiments to **benchmark V2T models** and evaluate the performance of our proposed model LOVE and LOVE-Reward strategy.

### 5.1. Experiment Setup

To evaluate the correlation between the predicted scores and the ground-truth MOSs, we utilize three evaluation criteria: Spearman Rank (SRCC), Pearson Linear (PLCC), and Kendall's Rank (KRCC) Correlation Coefficient. We compute correlations at two levels: **instance-level and model-level**. Instance-level evaluates how well the metric aligns with human ratings for each individual video, while model-level evaluates how well the metric ranks all 30 T2V models

based on their average score performance. We load the pre-trained weights for zero-shot model inference. Using the same training and testing split (49,500:9,000) as our model, we fine-tune three of the LMM-based models for comparison. The training set contains AIGVs from 18 open-source T2V models, while the test set contains AIGVs from all 30 T2V models to test the metric scalability. The prompts used in the training and test sets are non-overlapping to ensure fair evaluation. The models are implemented with PyTorch and trained on a 40GB NVIDIA RTX A6000 GPU with batch size of 4. The initial learning rate is set to 1e-5 and decreased using the cosine annealing strategy with Adam optimizer $\beta_1 = 0.9$ and $\beta_2 = 0.999$.

### 5.2. Evaluation on the AIGVE-60K Database

As shown in Table 2, handcrafted metrics such as BPRI (Min et al., 2017) and QAC (Xue et al., 2013), show poor performance, indicating their features handcrafted mainly for natural scenes are ineffective for evaluating AIGVs. VBench (Huang et al., 2024)'s fragmented evaluation framework, which combines multiple detection-based and pretrained metrics in a piecemeal fashion dependent on arbitrary threshold settings, leads to inconsistent cross-dimensional results. Vision-language pre-training models such as BLIPScore (Li et al., 2022b) and VQAScore (Li

*Table 3.* Comparisons of the alignment between different metric results and human annotations in evaluating T2V model performance. We report the average scores and QA accuracy at model level. ♠close-source T2V models unseen in LOVE training, ♡open-source T2V models. △Evaluation performance of all 30 models, ★Zero-shot evaluation performance on 12 close-source T2V models.

| Dimension | Perception Score | | | | Correspondence Score | | | | Question Answering Accuracy (%) | | | | Overall Rank | |
|---|---|---|---|---|---|---|---|---|---|---|---|---|---|---|
| Models | Human | Ours | V-Aesthetic | DOVER | Human | Ours | FGA-BLIP2 | Grok2v | Human | Ours | Qwen2.5 (72B) | Gemini | Human | Ours |
| ♠Pixverse (AI, 2025) | 63.81 | 62.46 | 59.06 | 67.42 | 59.97 | 61.65 | 66.34 | 86.23 | 91.33 | 88.00 | 80.67 | 76.67 | 1 | 1 |
| ♠Wanxiang (Cloud, 2025) | 60.54 | 61.54 | 57.34 | 63.86 | 60.37 | 60.84 | 62.56 | 85.45 | 90.33 | 89.33 | 84.00 | 80.00 | 2 | 2 |
| ♠Hailuo (Team, 2025d) | 60.58 | 62.40 | 56.47 | 65.37 | 59.74 | 60.63 | 60.51 | 86.26 | 87.67 | 87.33 | 84.00 | 83.33 | 3 | 4 |
| ♠Jimeng (Team, 2025a) | 65.25 | 65.72 | 61.97 | 74.16 | 57.86 | 59.12 | 62.44 | 79.40 | 81.33 | 79.33 | 72.33 | 68.67 | 4 | 5 |
| ♠Sora (Team, 2025e) | 62.09 | 62.71 | 58.23 | 67.49 | 59.68 | 60.33 | 61.21 | 84.81 | 85.67 | 85.00 | 78.33 | 80.00 | 4 | 2 |
| ♠Hunyuan (Li et al., 2024d) | 58.81 | 61.45 | 54.48 | 66.06 | 57.25 | 59.29 | 60.32 | 81.30 | 79.67 | 79.67 | 75.00 | 72.67 | 6 | 6 |
| ♠Vidu1.5 (Team, 2025f) | 54.56 | 55.19 | 55.83 | 47.61 | 58.25 | 58.15 |  | 82.90 | 87.00 | 84.33 | 83.00 | 83.33 | 7 | 8 |
| ♠Gen3 (Runway, 2025) | 59.22 | 62.52 | 59.21 | 66.17 | 55.72 | 57.75 | 55.91 | 74.77 | 75.33 | 79.67 | 71.00 | 65.00 | 8 | 6 |
| ♠Kling (Team, 2025c) | 60.56 | 61.34 | 55.48 | 61.58 | 55.57 | 58.21 | 59.70 | 76.37 | 73.67 | 79.00 | 73.67 | 65.33 | 9 | 8 |
| ♠Genmo (Team, 2025b) | 57.66 | 59.56 | 59.20 | 62.13 | 53.78 | 56.69 | 57.65 | 73.39 | 75.67 | 77.00 | 71.33 | 66.67 | 10 | 14 |
| ♠ChatGLM (GLM et al., 2024) | 56.39 | 60.96 | 56.81 | 60.87 | 53.98 | 57.78 | 56.11 | 79.54 | 74.00 | 77.33 | 72.67 | 73.33 | 11 | 10 |
| ♠Xunfei (Team, 2025g) | 58.60 | 61.79 | 59.78 | 67.53 | 53.46 | 56.00 | 57.11 | 67.71 | 66.33 | 74.67 | 63.67 | 58.00 | 12 | 13 |
| ♡Pyramid (Jin et al., 2024) | 63.67 | 63.15 | 60.29 | 67.46 | 50.17 | 56.35 | 55.35 | 67.97 | 50.17 | 71.33 | 63.67 | 57.33 | 13 | 12 |
| ♡Wan2.1 (Wang et al., 2025a) | 57.27 | 63.37 | 60.29 | 67.75 | 52.33 | 56.40 | 53.93 | 67.99 | 62.67 | 73.00 | 60.33 | 56.67 | 14 | 10 |
| ♡Allegro (Zhou et al., 2024) | 56.08 | 59.45 | 52.20 | 55.62 | 50.70 | 55.04 | 53.82 | 68.81 | 63.00 | 73.00 | 66.00 | 62.00 | 15 | 16 |
| ♡VideoCrafter2 (Chen et al., 2024a) | 48.11 | 56.99 | 55.24 | 49.67 | 51.07 | 57.16 | 58.86 | 70.99 | 65.67 | 73.00 | 78.67 | 59.67 | 16 | 15 |
| ♡CogVideo X1.5 (Yang et al., 2024) | 50.59 | 52.66 | 52.16 | 47.84 | 49.73 | 52.53 | 50.40 | 65.33 | 64.67 | 63.67 | 60.67 | 56.67 | 17 | 21 |
| ♡Animate (Xu et al., 2024) | 50.48 | 53.38 | 50.54 | 42.13 | 49.30 | 53.09 | 50.21 | 65.36 | 60.67 | 62.67 | 61.33 | 57.33 | 18 | 20 |
| ♡Lavie (Wang et al., 2023b) | 49.30 | 55.11 | 55.25 | 52.71 | 48.22 | 54.51 | 51.57 | 64.38 | 55.00 | 67.33 | 59.33 | 55.33 | 19 | 17 |
| ♡Hotshot-XL (Mullan et al., 2023) | 42.66 | 49.38 | 53.07 | 46.52 | 47.75 | 53.26 | 54.74 | 68.15 | 57.67 | 67.67 | 65.00 | 58.00 | 20 | 19 |
| ♡Latte (Ma et al., 2025) | 43.81 | 51.10 | 53.29 | 46.09 | 46.73 | 53.29 | 51.96 | 65.78 | 54.33 | 62.33 | 60.33 | 55.33 | 21 | 22 |
| ♡VideoCrafter1 (Chen et al., 2023) | 44.12 | 45.04 | 44.81 | 36.64 | 44.67 | 50.85 | 48.58 | 60.92 | 46.00 | 54.00 | 55.33 | 52.67 | 22 | 24 |
| ♡Text2Video-Zero (Khachatryan et al., 2023) | 40.53 | 46.03 | 57.55 | 48.45 | 44.89 | 51.69 | 51.39 | 61.94 | 48.67 | 60.67 | 55.67 | 57.67 | 23 | 23 |
| ♡NOVA (Deng et al., 2024) | 41.18 | 50.47 | 50.12 | 45.39 | 47.18 | 53.77 | 52.86 | 70.03 | 56.00 | 65.67 | 66.00 | 59.33 | 24 | 18 |
| ♡ModelScope (Wang et al., 2023a) | 38.00 | 41.22 | 46.15 | 29.92 | 43.73 | 48.99 | 45.54 | 59.99 | 47.33 | 51.33 | 58.00 | 53.00 | 25 | 26 |
| ♡Tune-A-Video (Wu et al., 2023b) | 35.41 | 40.87 | 53.75 | 42.34 | 42.69 | 49.81 | 49.46 | 59.85 | 43.00 | 57.67 | 57.33 | 56.33 | 26 | 25 |
| ♡LTX (HaCohen et al., 2024) | 40.11 | 42.70 | 47.09 | 46.13 | 41.28 | 48.08 | 42.79 | 55.83 | 37.00 | 48.00 | 57.00 | 46.67 | 27 | 27 |
| ♡LVDM (He et al., 2022) | 33.84 | 38.29 | 40.63 | 29.40 | 42.20 | 48.00 | 43.43 | 56.90 | 40.33 | 43.00 | 56.00 | 51.67 | 28 | 28 |
| ♡ZeroScope (Team, 2025h) | 30.08 | 37.58 | 42.92 | 34.89 | 34.69 | 43.76 | 38.99 | 42.13 | 22.00 | 32.67 | 45.33 | 37.67 | 29 | 29 |
| ♡LWM (Liu et al., 2024a) | 27.39 | 32.98 | 40.72 | 24.97 | 31.49 | 40.96 | 35.71 | 39.12 | 9.00 | 25.00 | 37.00 | 31.00 | 30 | 30 |
| △SRCC to human ↑ | - | 0.932 | 0.774 | 0.887 | - | 0.978 | 0.950 | 0.955 | - | 0.977 | 0.952 | 0.951 | - | 0.977 |
| △RMSE to human ↓ | - | 4.606 | 7.427 | 6.005 | - | 5.014 | 4.067 | 19.46 | - | 7.695 | 10.11 | 8.816 | - | 1.844 |
| ★SRCC to human ↑ | - | 0.825 | 0.266 | 0.650 | - | 0.944 | 0.811 | 0.909 | - | 0.904 | 0.828 | 0.807 | - | 0.942 |
| ★RMSE to human ↓ | - | 2.241 | 3.234 | 5.954 | - | 2.047 | 3.274 | 22.98 | - | 3.575 | 5.681 | 8.769 | - | 1.225 |

**Task -- Color** — T2V Models: Gen3, ModelScope, Pyramid. Prompt: *a white bag and a purple bed.*

| | Gen3 | ModelScope | Pyramid |
|---|---|---|---|
| Human | 69/70 ✔ | 34/33 ✘ | 66/68 ✔ |
| GPT-5.2 | 93/100 ✔ | 77/42 ✘ | 100/100 ✔ |
| LOVE (ours) | 67/68 ✔ | 34/37 ✘ | 65/67 ✔ |

**Task -- OCR** — T2V Models: Hailuo, ZeroScope, Sora. Prompt: *a video of phrase "Good luck"*

| | Hailuo | ZeroScope | Sora |
|---|---|---|---|
| Human | 69/69 ✔ | 35/31 ✘ | 69/67 ✔ |
| Gemini3-pro | 95/100 ✔ | 80/0 ✘ | 90/100 ✔ |
| LOVE (ours) | 66/64 ✔ | 36/33 ✘ | 67/64 ✔ |

**Task -- Position** — T2V Models: ChatGLM, Wan2.1, Jimeng. Prompt: *a cow below an airplane*

| | ChatGLM | Wan2.1 | Jimeng |
|---|---|---|---|
| Human | 47/61 ✔ | 51/31 ✘ | 62/65 ✔ |
| VideoLlama3 | 100/91 ✔ | 80/0 ✔ | 85/90 ✔ |
| LOVE (ours) | 49/63 ✔ | 60/39 ✘ | 61/67 ✔ |

*Figure 6.* Perception/Correspondence/QA prediction from different V2T interpretation methods compared to human annotation.

et al., 2024a) perform poorly in perception dimension due to their focus on T2V correspondence and overlook on perceptual quality. Although deep learning-based VQA methods achieve relatively better results, they still fall short in the T2V correspondence dimension. In contrast, LMM-based models achieve remarkable zero-shot generalization, especially in handling complex visual question-answering tasks, which highlights LMMs' inherent advantages in VQA tasks: they eliminate the need for VBench (Huang et al., 2024)'s fragile threshold tuning while overcoming traditional VQA's narrow specialization through unified multimodal understanding. The consistent performance of LMMs across all evaluation dimensions validates LMM's robustness as a comprehensive solution for AIGV assessment. Building upon these insights, we propose a LMM-based model that establishes an ***all-in-one*** evaluation framework and achieves superior performance in both score prediction and visual question answering, making it a more comprehensive method for evaluating AIGVs at both instance-level and model-level.

### 5.3. Evaluation on T2V Model Performance

We further conduct comparisons of the alignment between different metric results and human annotations in evaluating T2V model performance, as shown in Table 3. Our

model achieves the highest SRCC with human ratings and the lowest relative Root Mean Square Error (RMSE) in score differences. This demonstrates that our model is well-aligned with human judgment at the model level in assessing and ranking the performance of T2V models. Moreover, our model trained on 18 T2V models exhibits strong **scalability** in 12 unseen T2V model rank predictions. We also provide examples with model prediction scores at the instance level. As shown in Figure 6, LOVE generates scores that are more consistent with human annotations.

### 5.4. Zero-shot Cross-dataset Evaluation

Table 4 reveals LOVE's superior generalization capability. Our model is only trained on our AIGVE-60K dataset, and directly tested on other five datasets. Especially, compared to VQAScore trained on GenAI-Bench, our LOVE not only performs better on other four datasets, but also outperforms VQAScore on GenAI-Bench, which underscores LOVE's robustness and generality across diverse AIGV datasets.

### 5.5. Evaluation of Reward Methods

Table 5 and Figure 7 validate our proposed reward strategy. Notably, all reinforcement learning optimized methods surpass the pretrained baseline, validating the effectiveness

*Table 4.* Zero-shot cross-dataset correspondence performance on multiple benchmarks. *Refers to scores finetuned on the specific dataset.

| Method | AIGVE-60K | | FETV | | T2VQA-DB | | LGVQ | | AIGVQA-DB | | GenAI-Bench | |
|---|---|---|---|---|---|---|---|---|---|---|---|---|
| | SRCC | KRCC | SRCC | KRCC | SRCC | KRCC | SRCC | KRCC | SRCC | KRCC | SRCC | KRCC |
| CLIPScore (Hessel et al., 2021) | 0.229 | 0.143 | 0.607 | 0.498 | 0.049 | 0.033 | 0.446 | 0.301 | 0.152 | 0.101 | 0.536 | 0.180 |
| BLIPScore (Li et al., 2022b) | 0.211 | 0.127 | 0.616 | 0.505 | 0.174 | 0.118 | 0.455 | 0.319 | 0.181 | 0.122 | 0.546 | 0.201 |
| ImageReward (Xu et al., 2023) | **0.508** | **0.388** | 0.657 | 0.519 | 0.175 | 0.119 | 0.498 | 0.344 | 0.231 | 0.157 | 0.600 | 0.314 |
| PickScore (Kirstain et al., 2023) | 0.414 | 0.276 | 0.669 | 0.533 | 0.239 | 0.163 | 0.501 | 0.353 | 0.262 | 0.176 | 0.568 | 0.248 |
| HPSv2 (Wu et al., 2023c) | 0.499 | 0.379 | **0.686** | **0.540** | **0.243** | **0.168** | 0.504 | 0.357 | 0.229 | 0.153 | 0.515 | 0.137 |
| VQAScore (Li et al., 2024a) | 0.176 | 0.110 | 0.565 | 0.414 | 0.177 | 0.121 | **0.553** | **0.394** | **0.444** | **0.307** | **0.632*** | **0.382*** |
| **LOVE (Ours)** | **0.747*** | **0.560*** | **0.724** | **0.555** | **0.363** | **0.252** | **0.602** | **0.438** | **0.552** | **0.400** | **0.682** | **0.517** |

*Table 5.* Performance comparison of various reward methods built upon GRPO in terms of different reward evaluation metrics.

| Reward Method \ Evaluation Method | LOVE | | CLIPScore (Hessel et al., 2021) | PickScore (Kirstain et al., 2023) | ImageReward (Xu et al., 2023) | HPSv2 (Wu et al., 2023c) |
|---|---|---|---|---|---|---|
| | Perception | Correspondence | | | | |
| Pretrained (Wan2.1) | 0.5118 | 0.5175 | 0.5670 | 0.7316 | 1.5576 | 0.2570 |
| CLIPScore+GRPO | 0.5142 | 0.5182 | 0.5687 | 0.7322 | 1.5576 | 0.2570 |
| PickScore+GRPO | 0.5109 | 0.5172 | 0.5731 | 0.7324 | 1.5744 | 0.2571 |
| ImageReward+GRPO | 0.5106 | 0.5187 | 0.5700 | 0.7310 | 1.5925 | 0.2567 |
| HPSv2+GRPO | 0.5091 | 0.5181 | 0.5672 | 0.7323 | 1.5910 | 0.2571 |
| LOVE-Perception (Ours) | **0.5192** | 0.5211 | 0.5712 | 0.7323 | **1.6132** | 0.2594 |
| LOVE-Correspondence (Ours) | 0.5150 | **0.5256** | **0.5803** | **0.7395** | 1.6018 | **0.2632** |
| **LOVE-Reward (Ours)** | **0.5213** | **0.5268** | **0.5815** | **0.7406** | **1.6250** | **0.2643** |

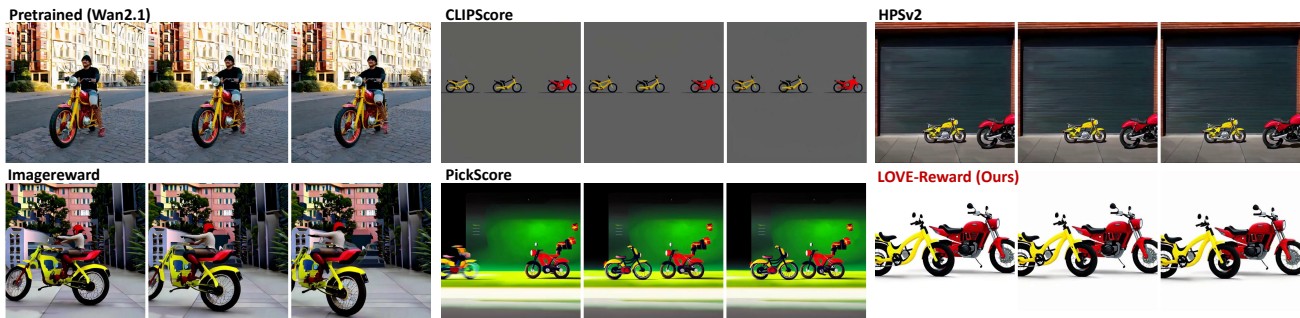

*Figure 7.* Video generation examples after finetuning by different reward methods using prompt: "*a yellow bicycle and a red motorcycle*".

*Table 6.* Ablation study on the temporal and quality-level features, LoRA strategy of LOVE.

| | Feature & Strategy | | | | Perception (ours) | | | Correspondence (ours) | | | QA (ours) | GenAI-Bench (Li et al., 2024a) | | |
|---|---|---|---|---|---|---|---|---|---|---|---|---|---|---|
| No. | Temporal | Quality level | Vision$_{r=16}$ | LLM$_{r=16}$ | SRCC | PLCC | KRCC | SRCC | PLCC | KRCC | Acc | SRCC | PLCC | KRCC |
| (1) | | | ✔ | ✔ | 0.642 | 0.658 | 0.458 | 0.597 | 0.612 | 0.397 | 78.4% | 0.550 | 0.553 | 0.412 |
| (2) | ✔ | ✔ | | | 0.756 | 0.781 | 0.618 | 0.723 | 0.735 | 0.584 | 65.8% | 0.652 | 0.663 | 0.498 |
| (3) | ✔ | ✔ | | ✔ | 0.779 | 0.808 | 0.589 | 0.742 | 0.760 | 0.549 | 77.3% | 0.676 | 0.689 | 0.509 |
| (4) | ✔ | ✔ | ✔ | | 0.781 | 0.813 | 0.593 | 0.739 | 0.758 | 0.552 | 76.8% | 0.674 | 0.690 | 0.506 |
| (5) | | ✔ | ✔ | ✔ | 0.734 | 0.762 | 0.601 | 0.737 | 0.759 | 0.548 | 78.4% | 0.673 | 0.694 | 0.501 |
| (6) | ✔ | ✔ | ✔ | ✔ | **0.793** | **0.826** | **0.602** | **0.747** | **0.766** | **0.560** | **78.7%** | **0.682** | **0.701** | **0.517** |

of reward-guided fine-tuning with the GRPO (Liu et al., 2025; Shao et al., 2024) framework. Our approach consistently outperforms all baseline methods across all evaluation metrics. The last three rows in Table 5 further confirms the complementary roles of the two dimensions: LOVE-Perception specializes in enhancing visual quality aspects, while LOVE-Correspondence focuses on improving text-video alignment. The combined LOVE-Reward delivers the most comprehensive improvements.

## 5.6. Ablation Study

We do extensive ablation investigations, as shown in Table 6. Our analysis reveals three main findings. First, experiments (1) and (5) demonstrate the effectiveness of text-defined quality-level initialization in model performance. Second, we confirm the notable performance improvements brought by LoRA fine-tuning strategy through experiments (2)–(4). Third, the efficacy of the temporal features is validated by experiments (5) and (6).

## 6. Conclusion

In this paper, we introduce **AIGVE-60K**, the largest AIGV evaluation dataset to date, consisting of 58,500 videos generated by 30 T2V models across 20 task-specific challenges and 2.6M subjective ratings. Based on AIGVE-60K, we benchmark and evaluate both the generation ability of T2V models and the V2T interpretation ability of LMMs. We also propose **LOVE**, a LMM-based evaluation model to achieve AIGV perceptual quality evaluation and T2V correspondence attribution in terms of both instance level and model level. Furthermore, we present **LOVE-Reward**, which effectively utilizes the multi-dimensional evaluation of LOVE to optimize T2V model generation through reinforcement learning. Extensive experiments demonstrate that LOVE achieves state-of-the-art performance, while LOVE-Reward consistently enhances video generation performance across multiple metrics, highlighting the significance of both the AIGVE-60K dataset and the proposed metrics.

## Acknowledgements

This work was supported in part by the National Natural Science Foundation of China under Grants 62522116, 62271312, 62132006, 62401365, 62225112, and U24A20220, in part by STCSM under Grant 22DZ2229005, and in part by the China Postdoctoral Science Foundation under Grants BX20250411 and 2025M773473.

## Impact Statement

We discuss how our work can be applied to benefit the community through four key contributions. **Firstly**, we resolve the long-standing limitation of evaluation subjectivity and scalability by constructing **AIGVE-60K**, the **largest** AIGV evaluation dataset with 58,500 videos and 2.6M human annotations explicitly disentangled into *perceptual quality* (visual artifacts, temporal consistency), *text-video correspondence* (prompt-video alignment), and *task-specific accuracy*. Unlike prior datasets with coarse merged scores (Liu et al., 2024c) or narrow expert judgments (Table 1), our fine-grained annotations eliminate hidden bias in human preference modeling while supporting both instance-level and model-level analysis. **Secondly**, we confront the methodological fragmentation in current evaluation ecosystems where isolated metrics like FVD (Unterthiner et al., 2018) lack prompt awareness and VBench's threshold-dependent pipelines (Huang et al., 2024) introduce inconsistent standards. Our **LOVE** framework establishes ethical transparency through an *all-in-one* LMM architecture with dual vision-temporal encoders and instruction tuning, unifying perceptual/correspondence/accuracy evaluation under reproducible criteria without arbitrary thresholds. **Thirdly**, we pioneer *bidirectional benchmarking* that benchmarking and evaluating 30 T2V video generation models and 48 V2T interpretation models on shared standards. By opensourcing annotations and maintaining minimal baseline tuning (LoRA adapters with 2-epoch training), we ensure community accessibility for: (1) engaging more T2V video generation models and V2T interpretation models for comparison, (2) developing more effective models based on our framework and training strategies. **Finally**, our proposed method LOVE-Reward bridgs the gaps from evaluation to generation. LOVE-Reward, which effectively translates the multi-dimensional evaluation capabilities of LOVE into a reward mechanism that significantly enhances T2V model generation through reinforcement learning. By providing fine-grained, multi-aspect feedback, LOVE-Reward enables models to progressively refine their outputs, leading to measurable performance gains in video quality, temporal coherence, and textual alignment.

We detail the ethical issues that may emerge during the dataset collection process. All participants in the subjective evaluation are clearly informed of the contents in our experiments. Specifically, we addressed the ethical problems by getting a written and informed permission from each person featured in the dataset stating that they approved their subjective ratings being used for non-commercial research, thus equipping it with such legal and ethical qualities. The experiments do not contain any visually improper or NSFW content (both *textual* and *visual*), because we used extensive manual review during the AIGV generation stage. Each image in our dataset has at least 15 annotators. All postgraduate students are in related fields from our laboratory and contributed to this process over 2 months, dedicating 3–4 hours daily to annotation tasks. We grouped the 58,500 analyzed videos into 30 sessions due to their vast volume. Each participant received $20 per session in accordance with the current ethical standard (Silberman et al., 2018; Otani et al., 2023). The experiment took more than two month to complete, with each participant contributing an average of 200 hours. All associated AIGVs and their corresponding prompts in the **AIGVE-60K** dataset are released under the **CC BY 4.0** license.

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
