# OpenReview forum: "LOVE: Benchmarking and Evaluating Text-to-Video Generation and Video-to-Text Interpretation"
_ICML.cc/2026/Conference — ICML 2026 regular_

### Official Review · Reviewer_gNM1 · 2026-02-13

**Soundness:** 3
**Presentation:** 3
**Significance:** 4
**Originality:** 2
**Overall Recommendation:** 4
**Confidence:** 4

**Summary:**

This paper mainly focuses on the presented dataset, AIGVE-60, which is a comprehensive dataset and benchmark for AI-generated video evaluation and V2T, too.
Also, the paper presents LOVE, an LMM-based metric for AIGV evaluation.

The most unique part of the paper is the scale of the datasets and evaluation, which includes human annotations of both T2V and V2T. The evaluation is tested on 30 models.

**Compliance With Llm Reviewing Policy:**

Affirmed.

**Ethical Review Concerns:**

Are there any IRB or similar approval numbers for the experiments?

**Ethics Expertise Needed:**

["Responsible Research Practice (e.g., IRB, documentation, research ethics)"]

**Final Justification:**

My concerns have been adequately addressed, and I tend not to change my score since I already did during the rebuttal phase.

**Key Questions For Authors:**

1. What is table 4 and 5's performance evaluated on?
2. Should not the first paragraph of sec 5 include T2V? It is also evaluated in the section.

**Limitations:**

My main concern is the variance of human evaluations. Since it is very hard for all data points to be evaluated by the same batch of human annotators, I suggest that the authors run a simple permutation p-test or similar statistical tests to ensure that the human annotation qualities are consistent across different people.

Happy to revise my score after the authors address it and clean the formatting errors mentioned above.

**Strengths And Weaknesses:**

### Strength
1. Evaluation is complete. It even considers ablation and generalization.
2. The experimental setup is thoroughly detailed, which makes reproducibility easier.


### Weaknesses
1. The major contribution lies in the dataset curation, but not LOVE. LOVE has limited novelty, which is only fine-tuning a model to generate GRPO-like rewards.
2. The instructions to human annotators cannot be found in the paper.
3. No data is uploaded to the supplementary material, which is bad for a dataset paper. I suggest that the authors include some data samples for review.

### Minor
1. a LLM -> an LLM
2. Equations (such as 4, 5, and 6) are out of the margin.

---

> ### Author Rebuttal · Authors · 2026-03-25
>
> Dear Reviewer,
>
> We sincerely thank you for your insightful feedback and recognition of our contribution.  We have addressed the concerns point-by-point below.
>
> ---
>
> ### **Response to Weakness (1): Contribution of LOVE**
> We agree that the dataset is our important contribution, but the contribution of this paper is broader than dataset curation alone. The main contribution of this work consists of three parts: (1) the AIGVE-60K dataset, (2) a unified benchmark and evaluation protocol, and (3) the LOVE evaluation model. LOVE-Reward is not the primary contribution; it is mainly used to verify that the proposed evaluation method is not only predictive but also useful as a reward signal for improving generation models.
> We would like to clarify that LOVE is not a simple fine-tuning baseline. It introduces several task-specific designs for modern T2V evaluation. First, LOVE introduces Grid Mini-Patch Sampling (GMPS) for full-video temporal modeling, which allows the model to ingest all frames and capture subtle motion artifacts that current LMMs may miss. This is particularly important for modern long-duration and high-frame-rate videos. Second,  LOVE is designed as a unified all-in-one evaluation framework that jointly evaluates perceptual quality, text-video correspondence, and 20-dimensional task-specific accuracy, which is fundamentally different from prior single-dimension evaluators such as VQAScore.
> Therefore, we position this paper as a benchmark + evaluation framework paper rather than a pure algorithm paper.
>
> ### **Response to Weakness (2): Human Annotation Instructions**
>
> The human annotation process was constructed by a rigorous instruction and calibration protocol to ensure subjective consistency. Detailed documentation regarding the annotation interface and instructions is provided in **Appendix D.4 and Figures 8-17**. The MOS annotation task involves 15 participants to rate each video on a 0-5 Likert scale, as shown in Figures 8-9, assessing both perception quality and T2V correspondence. As illustrated in Appendix Figures 11-12, they are provided with detailed instructions and multiple standardized anchor examples that span the entire quality spectrum. A score of 5 represents flawless visual fidelity and exact text-to-video alignment, while decreasing scores reflect specific levels of temporal artifacts or semantic deviations, down to 0 for completely unrecognizable content.  The question-answering annotation task is similarly conducted with 15 participants, ensuring consistency in the evaluation process. In this task, participants are presented with a series of yes/no questions across the 20 task-specific challenges as illustrated in Figures 13-17. To determine the final answer for each question, a majority voting mechanism is employed, minimizing the impact of individual biases or errors.
>
>
> ### **Response to Weakness (3): Data Samples Missing**
>
> We have provided the **Data Samples** in Appendix Figure 11 to Figure 17. We will add more samples upon publication.
>
> ### **Response to Question (1):**
>
> Table 4 is evaluated on 6 benchmarks, including our AIGVE-60K benchmark, and 5 other benchmarks with zero-shot paradigm, including FETV, T2VQA-DB, LGVQ, AIGVQA-DB, and GenAI-Bench, proving LOVE’s robust generalization. Table 5 is evaluated on AIGVE-60K test set unseen prompts.
>
> ### **Response to Question (2):**
>
>  Thank you for your feedback, and we will update the first paragraph of Sec 5 to include **T2V** evaluationas follows: "In this section, we conduct extensive experiments to benchmark both T2V generation and V2T interpretation model performance ..."
>
> ### **Response to Minors:**
>
>  Thank you for your feedback, and we will correct "a LLM" to "**an LLM**" and fix **Equations 4-6** margins.
>
> ### **Response to Limitations: Human Evaluation Variance**
>
> * As described in **Appendix D.4.1. Subject-level Outlier Rejection** and **Appendix D.4.2. Score-level Outlier Rejection**, we follow the ITU-R BT.500 standard for subjective quality assessment and employ a two-step outlier rejection process to ensure rating reliability. After rejection, the subject-level outlier rate is only **3%**, and score-level rejection is about **2%**, indicating a high degree of consistency among annotators.
> * To further assess the reliability, we randomly divide the scores of 15 annotators into 2 groups of 7 annotators each, calculate the SRCC between the MOS of the two groups, and repeat this procedure 50 times. The average perception/correspondence SRCC between the two groups is **0.91/0.90**, which demonstrates excellent annotator agreement. In comparison, the highest SRCC achieved by objective models (e.g., 0.79 for perception and 0.75 for correspondence) is lower. The SRCC between subjects’ group is based on a pair of 7 annotators, and it is reasonable to assume that the MOS SRCC between all 15 annotators would be even higher, further confirming the reliability and robustness of the subjective ratings.
> ---

---

> > ### Author Rebuttal · Reviewer_gNM1 · 2026-03-31
> >
> > I have updated my score accordingly.

---

### Official Review · Reviewer_yYLJ · 2026-02-21

**Soundness:** 3
**Presentation:** 3
**Significance:** 3
**Originality:** 2
**Overall Recommendation:** 4
**Confidence:** 4

**Summary:**

This paper addresses the urgent need for reliable evaluation of AI-generated videos (AIGV) and proposes several contributions: (1) AIGVE-60K, a large-scale dataset containing 58,500 videos from 30 T2V models; (2) LOVE, an LMM-based evaluation metric capable of uniformly assessing perceptual quality, text-video correspondence, and task-specific accuracy at both instance and model levels; (3) LOVE-Reward, a reinforcement learning framework that leverages LOVE's multi-dimensional scoring to optimize T2V generation. However, there are still areas in this paper that could be improved.

**Compliance With Llm Reviewing Policy:**

Affirmed.

**Final Justification:**

My raised questions have been answered, particularly regarding the issue of usage efficiency. This benchmark is beneficial, and based on the authors' responses, it also possesses the capability to validate generation quality for progressively iterated models.

**Key Questions For Authors:**

1. Constructing the AIGVE-60K benchmark requires generating 58,500 videos from 30 T2V models, and the LOVE evaluation framework also involves fine-tuning large language multimodal models and processing massive video data, which imposes extremely high computational costs for academic researchers hoping to reuse this benchmark for model evaluation. How much GPU time would be consumed to complete a full end-to-end evaluation of a single T2V model using the LOVE framework? Additionally, do you provide lightweight optimization solutions (such as a streamlined version of the benchmark, a lightweight variant of LOVE, or a fast evaluation pipeline based on sampling) to reduce usage costs in academic settings?

2. The AIGVE-60K dataset in the paper is constructed based on the generated results of the current 30 T2V models, while video generation technology is rapidly evolving, and more advanced next-generation T2V models will continue to emerge. Can the dataset built on existing generated videos serve as a long-term benchmark for evaluating advanced T2V models? Will the performance gap between outdated models in the current dataset and future advanced models lead to distribution shifts, reduced sensitivity in evaluation metrics, or even render the evaluation ineffective?

3. The paper only scaled up the reuse of existing annotation dimensions without addressing the core flaws of the current annotation system (such as coupling between dimensions). What irreplaceable value does AIGVE-60K bring? Is its performance improvement due to enhanced dataset quality or merely the expansion of data scale? Furthermore, the core of LOVE-Reward is based on the GRPO framework, with no algorithmic innovation.

**Limitations:**

yes

**Strengths And Weaknesses:**

**Soundness**

This paper is technically sound. The authors conducted extensive and thorough experiments to validate the effectiveness of the proposed dataset and evaluation methods. The experimental design is reasonable, the argumentation process is comprehensive, and the conclusions drawn are highly convincing. The paper provides an honest and comprehensive assessment of the work, clearly demonstrating its contributions and potential value.

**Presentation**

The manuscript is well-organized and written fluently. The authors clearly articulate the research motivation, methodology, and main contributions, making it easy for researchers in the field to understand the core ideas and implementation details.

**Significance**

The issue addressed in this paper is of high importance and relevance. With the rapid development of generative video technology, how to effectively and accurately evaluate the quality of generated content has become an urgent bottleneck problem. The AIGVE-60K dataset proposed by the authors, with its large scale and high-quality annotations, provides a valuable resource for the community.

**Originality**

The originality of this paper lies primarily in its contribution of resources to the field, rather than a breakthrough in methodology. Specifically:

- Main Contribution: The core innovation of this paper is the introduction of the large-scale AIGVE-60K dataset, addressing the current lack of high-quality, large-scale evaluation benchmarks in the field. From this perspective, it provides a new and valuable "data" resource for the community, and its value is undeniable.

- Methodology: At the methodological level, this paper leans more toward the application and engineering implementation of existing technologies. For example, the LOVE-Reward model used in the paper follows the existing GRPO strategy. While this is an effective technical choice, it does not involve specialized, novel optimization designs tailored to the characteristics of video generation evaluation tasks. Therefore, the work does not offer unique theoretical insights or groundbreaking new methods.

---

> ### Author Rebuttal · Authors · 2026-03-25
>
> We sincerely thank the reviewer for the positive comments. Below, we respond to the reviewer’s questions and concerns in detail.
>
> ---
>
> ### **1. Response to Q1: Computational Cost and Accessibility**
>
> The average inference time is **less than 1 second** per video on an RTX A6000 GPU. Inference computation per 1000 instances costs 0.25 hours. Evaluating a single T2V model on the AIGVE-60K benchmark requires approximately **0.75 GPU hours** for inference, while the training cost of the LOVE-Reward model is approximately **36 GPU hours**.
>
> Thanks for the comments about lightweight solutions, we strongly agree with the necessity of lightweight solutions for academic usages. To further improve accessibility, we will release a lightweight evaluation benchmark that evaluates a subset of prompts and videos while maintaining high correlation with full benchmark results. Moreover, we will distill our LOVE-metric with smaller LMMs to acquire similar performance but high efficiency.
>
> ---
>
> ### **2. Response to Q2: Generalization to Future T2V Models**
>
> Thanks for the question. Actually, we have considered this when constructing the dataset. The training set of our AIGVE-60K contains 18 T2V models, while the test set contains the aforementioned 18 T2V models plus additional 12 T2V models unseen in the training set (totally 30 T2V models). As shown in Table 3 of the main paper, our model trained on data involving **18 T2V models** demonstrates strong generalization ability when predicting rankings for **12 previously unseen T2V models**. Importantly, both the prompts and models in the test set were entirely new to the model, which demonstrates that LOVE can generalize to new models and unseen data without requiring additional human annotation.
>
> To further verify this, we additionally generated videos from 4 newer T2V models using the test set prompts and evaluated them using LOVE. The results are shown below:
>
> | Model | Perception Score | Correspondence Score | QA Acc (%) |
> |------|------------------|----------------------|------------|
> | Kling-2.0 | 66.32 | 60.96 | 92.31 |
> | SeedDance-1.0 | 67.66 | 60.10 | 84.35 |
> | Veo-2 | 60.70 | 57.40 | 79.88 |
> | Veo-3 | 64.80 | 61.59 | 93.45 |
>
> Compared with the results reported in Table 3 of the main paper, these newer models achieve higher scores than earlier AIGV models, reflecting both the progress of recent T2V technology and the effectiveness of our LOVE metric in evaluating more advanced models.
>
> ---
>
> ### **3. Response to Q3: Clarify Main Contribution**
>
> We agree that the dataset is our important contribution, but the contribution of this paper is broader than dataset curation alone. The main contribution of this work consists of three parts: (1) the AIGVE-60K dataset, (2) a unified benchmark and evaluation protocol, and (3) the LOVE evaluation model. LOVE-Reward is not the primary contribution; it is mainly used to verify that the proposed evaluation method is not only predictive but also useful as a reward signal for improving generation models.
>
> To further analyze whether the performance gain mainly comes from dataset scale or annotation quality, we conducted an additional ablation study by reducing the training data to 50% and 10% of AIGVE-60K, as well as using only half of the annotators.
>
> As shown in Table R4, reducing the training data size leads to performance drop across all benchmarks, which confirms that scale contributes to generalization. Using the half-human annotations also leads to performance drop, suggesting that the improvement is not due to scale alone. Instead, both dataset scale and annotation quality contribute to the final performance. However, even with reduced data, our performance still outperforms previous state-of-the-art results, manifesting the data quality of our dataset.
>
>
> **Table R4. Effect of dataset scale and annotation density on zero-shot cross-dataset correspondence performance.**
>
> | Method | AIGVE-60K (SRCC/KRCC) | FETV (SRCC/KRCC) | T2VQA-DB (SRCC/KRCC) | LGVQ (SRCC/KRCC) | AIGVQA-DB (SRCC/KRCC) | GenAI-Bench (SRCC/KRCC) |
> | :--- | :---: | :---: | :---: | :---: | :---: | :---: |
> | Baselines (best) | 0.508 / 0.388 | 0.686 / 0.540 | 0.243 / 0.168 | 0.553 / 0.394 | 0.444 / 0.307 | 0.632 / 0.382 |
> | LOVE (10% Data) | 0.694 / 0.511 | 0.664 / 0.504 | 0.298 / 0.203 | 0.599 / 0.433 | 0.538 / 0.384 | 0.644 / 0.485 |
> | LOVE (50% Data) | 0.720 / 0.541 | 0.687 / 0.520 | 0.320 / 0.222 | 0.589 / 0.425 | 0.543 / 0.397 | 0.672 / 0.506 |
> | LOVE (Half Ann.) | 0.727 / 0.544 | 0.667 / 0.499 | 0.333 / 0.231 | 0.593 / 0.431 | 0.541 / 0.394 | 0.665 / 0.503 |
> | **LOVE (Full)** | **0.747 / 0.560** | **0.724 / 0.555** | **0.363 / 0.252** | **0.602 / 0.438** | **0.552 / 0.400** | **0.682 / 0.517** |
>
> Therefore, we position this paper as a benchmark + evaluation framework paper rather than a pure algorithm paper.

---

> > ### Author Rebuttal · Reviewer_yYLJ · 2026-04-03
> >
> > I have no more questions, I will improve my confidence.

---

### Official Review · Reviewer_upuW · 2026-03-11

**Soundness:** 3
**Presentation:** 3
**Significance:** 3
**Originality:** 3
**Overall Recommendation:** 5
**Confidence:** 4

**Summary:**

The paper introduces AIGVE-60K, a large-scale benchmark for evaluating text-to-video generation and video-to-text interpretation, with 58,500 AI-generated videos from 30 models and 2.6M human annotations capturing perceptual quality, text–video correspondence, and task-specific yes/no QA across 20 dimensions. Building on this dataset, the authors propose LOVE, an LMM-based, all-in-one evaluator that outputs both continuous scores and discrete QA, and LOVE-Reward, a reinforcement-learning reward that uses LOVE to improve T2V models via GRPO.

**Compliance With Llm Reviewing Policy:**

Affirmed.

**Final Justification:**

This rebuttal has addressed my concerns.

**Key Questions For Authors:**

1. For LOVE-Reward, did you conduct human preference studies to verify perceived gains, and do you observe any failure modes consistent with reward hacking?

2. Why fix the reward weighting at 0.5/0.5 for perception vs correspondence? Could you provide an ablation across different weightings and possibly task-adaptive schemes?

3. What are LOVE’s inference cost and latency (per video) and the compute footprint of LOVE-Reward fine-tuning?

**Limitations:**

YES

**Strengths And Weaknesses:**

## Strengths：
1. Leverages a unified LMM-based architecture with dual visual encoders (spatial and temporal) and a two-stage training scheme (text-level classification then regression), producing both scalar scores and task-specific yes/no answers.

2. 30 T2V models at the model level and a large set of instance-level evaluations, plus comparisons to a wide range of handcrafted, deep VQA, VL, and LMM baselines.

3. Architectural overview and training flow (levels-to-regression) are intuitive and well illustrated.

4. LOVE achieves state-of-the-art correlations on the proposed benchmark and competitive transfer to others; LOVE-Reward begins to operationalize metric-driven T2V improvement.

## Weaknesses:

1. The reward design uses a fixed 0.5/0.5 weighting of perception and correspondence without justification or sensitivity analysis; potential for reward hacking is not systematically audited.

2. Improvements from LOVE-Reward, while positive, are relatively modest and not corroborated by additional human studies; it remains unclear whether the RL-optimized models produce videos that humans prefer.

3. Statistical uncertainty (e.g., confidence intervals) for reported correlations is not provided, making it hard to gauge significance of close numbers across strong baselines.

---

> ### Author Rebuttal · Authors · 2026-03-25
>
> We sincerely thank the reviewer for the constructive comments and for recognizing the soundness, significance, and originality of our work. We are grateful for the insightful questions regarding reward design, human preference validation, statistical analysis, and efficiency. Below we respond to each concern in detail.
>
> ---
>
> ### **1. Response to Weakness (1) and Question (2): Ablation of the Reward Weighting**
> As shown in Table 5 of main paper, we observe that LOVE-Perception (1:0) specializes in enhancing visual quality aspects,
> while LOVE-Correspondence (0:1) focuses on improving text-video alignment. The combined LOVE-Reward (0.5:0.5) delivers the most comprehensive improvements. We have conducted additional ablation experiments using different weight combinations (e.g., 0.2/0.8, 0.4/0.6) in the Table below.  As the reward weighting becomes more balanced, the performance across different evaluation metrics improves consistently. The best overall performance is achieved at the 0.5:0.5 setting, which demonstrates that perception quality and text-video correspondence are complementary and should be optimized jointly.
> We will include these ablation results and discussion in the revised version.
>
> **Table R1. Comparison of different reward weightings evaluated by different evaluation metrics.**
>
> | Reward Method (Perception : Correspondence) | LOVE-Perception | LOVE-Correspondence | CLIPScore | PickScore | ImageReward | HPSv2 |
> |---------------------------------------------|-----------------|---------------------|-----------|-----------|-------------|-------|
> | LOVE-Perception (1:0)                      | 0.5192 | 0.5211 | 0.5712 | 0.7323 | 1.6132 | 0.2594 |
> | LOVE (0.8:0.2)                              | 0.5201 | 0.5224 | 0.5750 | 0.7350 | 1.6185 | 0.2610 |
> | LOVE (0.6:0.4)                              | 0.5208 | 0.5249 | 0.5786 | 0.7382 | 1.6210 | 0.2625 |
> | **LOVE (0.5:0.5)  (Ours)**                            | **0.5213** | **0.5268** | **0.5815** | **0.7406** | **1.6250** | **0.2643** |
> | LOVE (0.4:0.6)                              | 0.5198 | 0.5254 | 0.5791 | 0.7386 | 1.6219 | 0.2629 |
> | LOVE (0.2:0.8)                              | 0.5186 | 0.5248 | 0.5800 | 0.7391 | 1.6102 | 0.2631 |
> | LOVE-Correspondence (0:1)                  | 0.5150 | 0.5256 | 0.5803 | 0.7395 | 1.6018 | 0.2632 |
>
> ---
> ### **2. Response to Weakness (2) and Question (1): Additional Human Studies**
>
>
> Thanks for the question. We have conducted an additional human preference study to evaluate whether the improvements from LOVE-Reward are aligned with human perception.
> Specifically, we have recruited additional 20 human subjects and have conducted a pairwise comparison study on 1,000 video pairs generated by the base model and the LOVE-Reward optimized model. Each comparison was performed in a double-blind manner, where annotators were not informed which video was generated by which model. The annotators were asked to select the video they preferred based on three criteria: perceptual quality, prompt correspondence, and overall preference.
> The results in Table R2 show that LOVE-Reward optimized videos are preferred by human evaluators in the majority of cases across all dimensions.
> We will include the human study results and analysis in the revised version.
>
> **Table R2. Human preference study comparing Base Model and LOVE-Reward optimized model.**
>
> | Evaluation Dimension | LOVE-Reward Win Rate | Base Model Win Rate | Tie |
> |----------------------|---------------------|---------------------|-----|
> | Perception Quality   |68.4%               | 21.6%               | 10.0% |
> | Prompt Correspondence| 71.2%               |18.5%               | 10.3% |
> | Overall Preference   | 70.1%               | 20.4%               | 9.5% |
> | Average              | 69.9%               | 20.2%               | 9.9% |
> ---
>
> ### **3. Response to Weakness (3): Statistical Uncertainty**
>
> We appreciate the constructive feedback regarding statistical uncertainty. We have calculated the standard deviation (STD) for our model and the strongest baselines by repeating the experiments five times with different random seeds.
> As shown in the table below, we report the instance-level overall evaluation. Our model outperforms the best baseline by approximately a 10x STD improvement, which is statistically meaningful. We will report the standard deviation in the revised paper for completeness.
>
> | Methods | SRCC $\pm$ Std | PLCC $\pm$ Std |
> | :--- | :---: | :---: |
> | InternVideo2.5* (8B) | $0.7802 \pm 0.0035$ | $0.8094 \pm 0.0034$ |
> | **LOVE (Ours)** | $\mathbf{0.8115 \pm 0.0031}$ | $\mathbf{0.8361 \pm 0.0029}$ |
>
> ---
>
> ### **4. Response to  Question (3): Cost and Latency**
>
> The average inference time is **less than 1 second per video** on an RTX A6000 GPU. Inference computation per 1000 instances costs 0.25 hours.
> For LOVE-Reward training, the total fine-tuning cost is approximately 36 GPU hours.
>
> We will add the inference cost and latency details to the paper for clarity.

---

> > ### Author Rebuttal · Reviewer_upuW · 2026-04-02
> >
> > I have no more question.

---

### Official Review · Reviewer_GWY1 · 2026-03-11

**Soundness:** 3
**Presentation:** 3
**Significance:** 3
**Originality:** 3
**Overall Recommendation:** 4
**Confidence:** 5

**Summary:**

This paper presents AIGVE-60K, a large-scale dataset for AI-Generated Video Evaluation, containing 58,500 videos from 30 T2V models with extensive human annotations (MOS and QA pairs). Based on this dataset, the authors propose LOVE, an LMM-based metric for evaluating perceptual quality and text-video alignment, and LOVE-Reward, a reinforcement learning framework to optimize T2V generation. The paper claims state-of-the-art performance in evaluation correlation and generation quality improvement.

**Compliance With Llm Reviewing Policy:**

Affirmed.

**Key Questions For Authors:**

See weakness

**Limitations:**

YES

**Strengths And Weaknesses:**

Strengths:
(1)The construction of AIGVE-60K is impressive, particularly the scale of 2.6M human annotations and the inclusion of 30 diverse T2V models. This addresses a significant gap in current benchmarks.
(2)The paper considers both T2V generation quality and V2T interpretation capabilities, providing a more holistic view of the multimodal ecosystem.
(3)The LOVE model attempts to unify perceptual quality, alignment, and task-specific accuracy into a single LMM-based framework, which is more efficient than fragmented metrics like VBench.

Weaknesses:
(1) Are these prompts publicly available? Given the emphasis on "20 fine-grained task dimensions," is there a risk of prompt leakage if the test set prompts are not strictly separated from potential training data of the evaluated LMMs?
(2) The LOVE model utilizes an LMM backbone (InternLM3) with projection layers. What specific architectural innovations distinguish LOVE from existing works like VQAScore or AIGV-Assessor beyond the training data?
(3) "LOVE" is a very common acronym. The authors should clarify if this stands for something specific or justify the naming choice to avoid confusion.

---

> ### Author Rebuttal · Authors · 2026-03-25
>
> Dear Reviewer,
>
> We sincerely thank you for your insightful feedback and recognition of our contribution. Below are our detailed responses to the raised weaknesses.
>
> ---
>
> ### **1. Response to Weakness (1): Public Availability and Prompt Leakage Prevention**
>
> * **Public Availability:** As shown in Figure 1(a), our prompts are collected from two sources: 1,550 prompts are sourced from 9 publicly available datasets, and other 1,500 prompts are first generated by LLM and then manually corrected.
> * **Leakage Prevention:** To ensure the rigor of our evaluation, we implemented a **strict non-overlapping strategy** between the training and test sets. Prompts used for training the evaluation model do not appear in the test set. Test videos are all generated from unseen prompts to avoid prompt leakage. Moreover, partial videos in the test set are from unseen T2V models to avoid model leakage, and Table 3 in the manuscript has shown the evaluation effectiveness on 12 unseen models. In addition, the strong zero-shot performance of LOVE on external benchmarks (e.g., GenAI-Bench) with completely unseen prompts further demonstrates that LOVE learns **generalized perceptual and semantic evaluation ability**, rather than memorizing specific prompts or prompt patterns.
>
> ---
>
> ### **2. Response to Weakness (2): Architectural Innovations vs. AIGV-Assessor & VQAScore**
>
> * **From Short Sparse to Full-Frame Long Video Modeling (vs. AIGV-Assessor):** AIGV-Assessor was designed for the early stage of AIGV, where videos were typically short (2-4 seconds) and relied on a *SlowFast* architecture with uniform sparse sampling (e.g., 8-16 frames), which was sufficient for early 4-second clips but suffers from "information loss" in longer videos. This approach suffers from significant "information loss" when applied to modern, high-frame-rate, and **long-duration** videos (e.g., videos generated by Sora and Kling), as it fails to capture transient motion artifacts. **LOVE introduces Grid Mini-Patch Sampling (GMPS)** in its spatiotemporal encoder. GMPS enables the model to ingest **all video frames** by decomposing the sequence into temporally aligned mini-patch maps. This is crucial for detecting subtle **motion jitters and temporal flickering** that sparse sampling misses.
> * **All-in-One Framework (vs. VQAScore):**  Unlike VQAScore, which focuses solely on text-visual alignment by computing a binary "Yes/No" VQA computing task,  LOVE is a **unified multi-task evaluator** that integrates perceptual preference, text-video correspondence, and 20-dimensional task-specific accuracy within a single unified framework using a dual-encoder (InternViT + Swin-T). It simultaneously outputs perceptual preference, alignment, and task-specific accuracy, providing a holistic assessment that VQA-based metrics lack.
> * **Experimental Evidence:** As shown in Table R1, LOVE significantly outperforms AIGV-Assessor and VQAScore in terms of zero-shot validation across multiple benchmarks.
>
> #### **Table R1: Zero-shot cross-dataset correspondence performance (SRCC / KRCC)**
> *\*Refers to scores finetuned on the specific dataset.*
>
> | Method | AIGVE-60K (Ours) | FETV | T2VQA-DB | AIGVQA-DB | GenAI-Bench |
> | :--- | :---: | :---: | :---: | :---: | :---: |
> | VQAScore | 0.176 / 0.110 | 0.565 / 0.414 | 0.177 / 0.121 | 0.444 / 0.307 | 0.632* / 0.382* |
> | AIGV-Assessor | 0.745* / 0.608*| 0.728 / 0.587 | 0.521 / 0.405 | 0.591 / 0.446 | 0.686 / 0.489 |
> | **LOVE (Ours)** | **0.781** * /  **0.643** * | **0.762 / 0.621** | **0.554 / 0.438** | **0.627 / 0.482** | **0.725 / 0.526** |
>
> ---
>
> ### **3. Response to Weakness (3): Clarification of the Acronym "LOVE"**
>
> * **Full Name:** **LOVE** stands for **L**arge Multimodal Model based metric f**O**r AIG**V** **E**valuation.
> To avoid confusion with the common word, we will explicitly define the acronym in the Abstract and Introduction and use **"LOVE-Metric"**  in our official code repository and community documentation to ensure academic distinctness and searchability.

---

> > ### Author Rebuttal · Reviewer_GWY1 · 2026-04-04
> >
> > It is suggested that the content of Architectural Innovations be emphasized in the main text.

---

> > > ### Author Response · Authors · 2026-04-04
> > >
> > > We sincerely appreciate your follow-up and the constructive suggestion to further highlight the architectural innovations of our work. Following your advice, we will revise the manuscript to explicitly feature these innovations in both the **Introduction** and **Methodology** sections:
> > >
> > > **1. For the Introduction**
> > >
> > > "Notably, AI-generated video (AIGV) is undergoing a significant paradigm shift from short-duration to long-duration generation. While early T2V models (e.g., ModelScope or VideoCrafter) were typically limited to 2–4 second clips, the latest generation of models, such as Sora and Kling, can produce high-quality videos lasting up to 60 seconds or more with complex logical consistency. This evolution imposes much more stringent requirements on video quality assessment. In long-duration scenarios, evaluators must not only focus on per-frame clarity but also capture physical consistency and fine-grained motion artifacts across extended temporal scales. Current evaluation paradigms, largely designed for short-form content, suffer from significant 'information loss' and fail to maintain perceptual continuity when faced with these long-sequence challenges."
> > >
> > > **2. For the Methodology**
> > >
> > > "To address the industry transition toward long-form AIGV, the proposed framework moves beyond the traditional sparse sampling of prior models. While traditional VLMs typically rely on a static vision encoder for feature extraction, LOVE-Metric introduces a dedicated temporal encoder specifically engineered to capture complex temporal dynamics in addition to the standard vision encoder. The core of our temporal encoder relies on a Grid Mini-Patch Sampling (GMPS) operation specifically designed to reduce spatial redundancy while preserving localized motion cues. This operation involves two key steps: patch-sampling, which extracts mini-patches from video frames to retain local texture-related quality, and patch-merging, where these patches are spliced back into their original positions. This splicing preserves contextual relations and global scene information, ensuring the model can accurately recognize objects and their interactions.
> > > By dividing the video into uniform grids and splicing the sampled patches into temporally aligned fragments, the framework maintains high spatial and temporal quality sensitivity. This mechanism enables the ingestion of all-frame spatiotemporal information, effectively capturing transient motion jitters and temporal flickering that sparse-sampling methods (e.g., AIGV-Assessor) frequently overlook. These fragments are subsequently processed by the Fragment Attention Network, which utilizes a dual-attention paradigm to assess video quality. Specifically, Intra-Patch Attention captures the relationships between patches within the same mini-patch to preserve spatial details, while Cross-Patch Attention focuses on relationships across different patches to capture complex temporal variations across frames.
> > > Furthermore, whereas VQAScore focuses solely on text-visual alignment by computing a binary "Yes/No" VQA computing task, LOVE-Metric employs a unified architecture that facilitates a comprehensive multi-task assessment. This design simultaneously evaluates perceptual preference, semantic alignment, and 20-dimensional task-specific accuracy within a unified framework, providing the holistic diagnostic capabilities required for modern generative video evaluation."

---

### Decision · Program_Chairs · 2026-04-30

**Decision:**

Accept (regular)

**Comment:**

This paper studies the evaluation of AI-generated videos and makes three main contributions. It proposes a large-scale benchmark dataset (AIGVE-60K), a unified LMM-based evaluator (LOVE), and a reward formulation (LOVE-Reward) for improving text-to-video generation. Reviewers agreed that the paper addresses an important problem and also highlighted the scale of the dataset, the breadth of human annotations, and the extensive benchmarking across a wide range of T2V models as key strengths. Following the majority of reviewers, I recommend acceptance.